# The Oral Transglutaminase 2 Inhibitor ZED1227 Accumulates in the Villous Enterocytes in Celiac Disease Patients during Gluten Challenge and Drug Treatment

**DOI:** 10.3390/ijms241310815

**Published:** 2023-06-28

**Authors:** Jorma Isola, Markku Mäki, Martin Hils, Ralf Pasternack, Keijo Viiri, Valeriia Dotsenko, Toni Montonen, Timo Zimmermann, Ralf Mohrbacher, Roland Greinwald, Detlef Schuppan

**Affiliations:** 1Jilab Inc., 33520 Tampere, Finland; 2Faculty of Medicine and Health Technology, Tampere University, 33014 Tampere, Finland; markku.maki@tuni.fi (M.M.); valeriia.dotsenko@tuni.fi (V.D.);; 3Zedira GmbH, Roesslerstrasse 83, 64293 Darmstadt, Germany; hils@zedira.com (M.H.); pasternack@zedira.com (R.P.); 4Dr. Falk Pharma Gmbh, 79108 Freiburg, Germany; timo.zimmermann@drfalkpharma.de (T.Z.); roland.greiwald@drfalkpharma.de (R.G.); 5Institute of Translational Immunology and Celiac Center, Medical Center, Johannes-Gutenberg University, 55099 Mainz, Germany; 6Division of Gastroenterology, Beth Israel Deaconess Medical Center, Harvard Medical School, Boston, MA 02115, USA

**Keywords:** celiac disease therapy, gliadin, gluten, enterocyte, brush border

## Abstract

The enzyme transglutaminase 2 (TG2) plays a key role in celiac disease (CeD) pathogenesis. Active TG2 is located mainly extracellularly in the lamina propria but also in the villous enterocytes of the duodenum. The TG2 inhibitor ZED1227 is a promising drug candidate for treating CeD and is designed to block the TG2-catalyzed deamidation and crosslinking of gliadin peptides. Our aim was to study the accumulation of ZED1227 after oral administration of the drug. We studied duodenal biopsies derived from a phase 2a clinical drug trial using an antibody that detects ZED1227 when bound to the catalytic center of TG2. Human epithelial organoids were studied in vitro for the effect of ZED1227 on the activity of TG2 using the 5-biotin-pentylamine assay. The ZED1227-TG2 complex was found mainly in the villous enterocytes in post-treatment biopsies. The signal of ZED1227-TG2 was strongest in the luminal epithelial brush border, while the intensity of the signal in the lamina propria was only ~20% of that in the villous enterocytes. No signal specific to ZED1227 could be detected in pretreatment biopsies or in biopsies from patients randomized to the placebo treatment arm. ZED1227-TG2 staining co-localized with total TG2 and native and deamidated gliadin peptides on the enterocyte luminal surface. Inhibition of TG2 activity by ZED1227 was demonstrated in epithelial organoids. Our findings suggest that active TG2 is present at the luminal side of the villous epithelium and that inhibition of TG2 activity by ZED1227 occurs already there before gliadin peptides enter the lamina propria.

## 1. Introduction

Celiac disease (CeD) is an autoimmune disorder in which dietary gluten causes a gradually developing inflammation, villous atrophy, and crypt hyperplasia in the mucosa of the small intestine. The enzyme transglutaminase 2 (TG2) plays a key role in celiac disease (CeD) pathogenesis, both as a CeD autoantigen and through the generation of immunogenic, deamidated gliadin peptide epitopes recognized by the T-cells [1]. TG2 is thought to localize mainly extracellularly in the lamina propria of the small bowel mucosa [2], where it is a target for humoral immunity [3]. In celiac disease, TG2 forms complexes with immunoglobulin A on the subepithelial basement membrane, forming the IgA deposits detectable by immunofluorescence microscopy [3].

In addition to its localization in the lamina propria, single-cell transcriptome and functional studies suggest that TG2 can be active also in or on villous enterocytes [4,5]. Studies using immunofluorescence and immunoelectron microscopy have also demonstrated the presence of TG2 protein on the luminal enterocyte surface, especially on the microvilli [6,7]. TG2 is mainly localized intracellularly in most cell types but can be released upon cellular stress or during necroapoptosis, where it becomes activated due to high extracellular calcium [7]. The estimated half-life of active TG2 in the presence of substrates such as gliadin that can be cross-linked or deamidated is short [8,9].

Inhibitors of TG2 have been suggested as promising new drug candidates for CeD, designed to block the deamidation of gliadin peptides, which is necessary for the initiation of robust gluten-induced T-cell activation and inflammation in patients with CeD [9,10]. Proof of concept of the efficacy of the oral TG2 inhibitor ZED1227 was recently obtained in a clinical phase 2a gluten challenge trial, where the drug significantly attenuated both gluten-induced small intestinal mucosal morphological deterioration as well as inflammation, and improved symptom and quality-of-life scores [11].

To better understand the mode-of-action of ZED1227, we developed a monoclonal antibody against ZED1227 and studied its accumulation relative to TG2 protein and gliadin peptides using immunofluorescence and confocal microscopy in duodenal biopsies from patients treated with ZED1227. Furthermore, we used human epithelial organoids [12] to obtain direct functional evidence of transglutaminase inhibition by ZED1227.

## 2. Results

### 2.1. Localization of TG2 and Gliadin Peptides

The mucosal localization of the transglutaminase 2 enzyme was studied using the monoclonal antibody CUB7402. Most of the fluorescence signal was localized in the lamina propria, but the same signal intensity was also found on the luminal surface of the villous epithelium (Figure 1A). Immunofluorescence staining of non-deamidated (native) and deamidated gliadin 33-mer peptide (using anti-gliadin antibody A161) showed accumulation also on the luminal surface of the villous enterocytes (Figure 1B). A hematoxylin-eosin-stained biopsy from a CeD patient is shown for reference in Appendix A.

### 2.2. Localization of the ZED1227

The co-localization of TG2 and gliadin peptides raised the question of whether the ZED1227-TG2 interaction takes place already in the enterocytes. The immunofluorescence signal using monoclonal antibody A083 results from the ZED1227-TG2 conjugate because free, unbound ZED1227 is washed away from the tissue and ZED1227 does not react with other proteins. The signal was detected in villous enterocytes already at low magnification in biopsies from patients who had received the last oral dose of ZED1227 already 24 h earlier (Figure 2A). The signal was cytoplasmic and concentrated on the luminal side cell membrane in the brush border area at higher magnification (Figure 2B,C). It is possible, if not likely, that the signal localized on the luminal brush border is extracellular. A lower-intensity signal was seen in the underlying lamina propria. No detectable fluorescence signal could be detected in the pretreatment biopsies from the same patients (Figure 2C,D) or in patients randomized to the placebo arm (not shown). Pretreatment of A083 with ZED1127 before application to the tissue sections completely abolished the fluorescence signal (Appendix A).

A comparison of the integrated A083 fluorescence signal intensities for ZED1227 from biopsies belonging to the 10 mg, 50 mg, and 100 mg ZED1227 dose patient groups showed a clear dose-dependent increase in signal intensity (Figure 3A). Since the majority of total TG2 was localized in the lamina propria, we quantitated the mean fluorescence signal intensities for ZED1227 separately in the lamina propria and enterocyte areas (Figure 3). Background-corrected fluorescence intensity of ZED1227 was on average 2.7 times stronger in the villous epithelium compared to the lamina propria in the 100 mg drug dose group (Figure 3). Western blots demonstrating the ZED1227-TG2 antibody specificity and technical staining controls are shown in Appendix A.

### 2.3. Co-Localization of TG2 Protein and ZED1227

Double immunofluorescence and confocal microscopy were used to study the co-localization of total TG2 and ZED1227 at high resolution (Figure 4). A quantitative co-localization algorithm (Imagej/FIJI, [13]) showed significant signal co-localization at the pixel level in the epithelial luminal surface and in the lamina propria (Figure 4).

### 2.4. Inhibition of TG2 by ZED1227 in a Lamina Propria-Free Intestinal Organoid Model

We next studied the expression and localization of TG2 and ZED1227 in human intestinal epithelial organoids that are devoid of lamina propria. The staining results of TG2 protein were similar to those obtained from clinical biopsies, with prominent deposition on the organoid epithelial cell surface (Figure 5A). Transcript levels of TG2 were determined by qRT-PCR. A substantial level of expression of TG2 mRNA was found, with a clear trend to increase once the organoid epithelium was differentiated towards mature villous epithelium by adding the ELR1 culture medium (Figure 5B). Administration of ZED1227 to the culture did not alter the mRNA expression of TG2. When ZED1227 was added to the culture medium, it was detected in the differentiated epithelial cells, similar to the pattern seen in clinical biopsies (Figure 5D). The effect of ZED1227 on TG2 activity was studied after adding the substrate 5-biotin pentylamine (5-BP). Staining intensity on the epithelial cell surface, reflecting TG2 activity, was decreased dose-dependently compared to buffer control after treatment with 0.002 or 0.03 mg/mL of ZED1227 (Figure 6).

## 3. Discussion

The efficacy of TG2 inhibitor ZED1227 to attenuate gluten-induced mucosal damage was previously reported in a proof-of-concept, randomized, double-blind, placebo-controlled, 6-week phase 2 trial in 160 patients with celiac disease in remission who were challenged with 3 g of gluten per day [11]. Here, our novel and unexpected finding was that the oral TG2 inhibitor ZED1227 concentrated mainly in the villous enterocytes and not in the lamina propria, where the majority of TG2 protein resides in the duodenal mucosa. The localization of the TG2 protein in the lamina propria, especially in the basement membrane, is extensively demonstrated in the literature [2,3]. Its localization on the luminal surface of the enterocytes has also been documented [6,7,13], but the role of the luminal TG2 in celiac disease pathogenesis has remained unclear. Our results from clinical biopsies and in vitro organoids suggest that pathogenic deamidation of gliadin peptides by TG2 and its inhibition by ZED1227 may occur in the villous epithelium and possibly to a lesser degree in the lamina propria, unlike previously assumed.

At the subcellular level, the orally administered TG2 inhibitor ZED1227 was trapped in the luminal brush border compartment, along with the partly digested immunogenic gliadin peptides detected by immunofluorescence staining. Despite the lack of direct evidence, most review articles suggest that gliadin peptide deamidation by TG2 occurs in the lamina propria since the lamina propria harbors the T cells that drive CeD pathogenesis [1]. However, TG2-mediated deamidation of gluten peptides was suggested to take place in the epithelial brush border already in 2002 ([13] Sollid 2002). Our results strengthen the hypothesis based on mouse studies that villous epithelial TG2 is a major driver of gluten deamidation in CeD [4,13]. Sollid et al. [14] concluded that if pathogenetic extracellular TG2 is mainly derived from the enterocytes, active TG2 will possibly also be shed towards the gut lumen, where it can react with gluten peptides before transepithelial transport to the lamina propria.

The disproportional distribution of TG2 protein and ZED1227-TG2 may have several underlying explanations. First, the half-life of TG2 in the extracellular space of the lamina propria is probably short due to high calcium concentrations compared to the usually low calcium in the intestinal lumen [8]. Second, the biopsies were taken 24 h post-dosage. Therefore, ZED1227 may already have been removed after binding to TG2 in the lamina propria, either by phagocytosis and/or rapid transport into the circulation [10]. Third, TG2 in the extracellular lamina propria may have a different tertiary structure, and it is partially covered by immunoglobulins, seen as IgA deposits near the basement membrane [2,3]. In this case, TG2 inhibitors may not be able to bind their target effectively.

Our results showed that ZED1227, due to covalent binding to TG2, is still detectable 24 h after the last drug dose in the villus epithelium. It is possible that we have missed a time point of maximal drug concentration in the lamina propria if TG2 turnover there is more rapid. Nonetheless, persistent TG2 activity and ZED1227 binding occur on the luminal side of the intestinal epithelium, indicating a relevant contribution of epithelial TG2 to CeD pathogenesis. This may have important implications for the treatment of CeD with oral TG2 inhibitors.

## 4. Materials and Methods

### 4.1. Patients and Biopsies

A randomized phase 2a study (CEC-3/CEL) investigating the efficacy and safety of the TG2 inhibitor ZED1227 (also known as TAK227) in CeD patients has been reported recently [11]. Distal duodenal biopsies were taken at baseline and after a 6-week gluten challenge (3 g of gluten per day). Patients were randomized to receive one daily capsule of placebo, 10 mg, 50 mg, or 100 mg of ZED1227 concurrent with the gluten challenge. Post-treatment biopsies were taken at an endoscopy performed 24 h after the last dose of the drug or placebo. The biopsies were fixed with Paxgene [15] and processed as paraffin blocks. Tissue sections (3 μm thick) of pre-and post-treatment biopsies from each patient were cut and placed on the same microscope slide to ensure the comparability of the staining results.

### 4.2. Antibodies

Anti-ZED1227. A mouse monoclonal antibody (clone A083, Zedira, Darmstadt, Germany) was raised against the ZED1227 analog conjugated with KLH (Keyhole Limpet Hemocyanin). Briefly, the Michael acceptor warhead was replaced by (L)-homocysteine, a non-proteinogenic thiol amino acid, thereby facilitating the nucleophilic coupling of the peptidomimetic compound to maleimide-activated KLH (Sigma, K0383, St. Louis, MO, USA). The conjugate was used to immunize mice, followed by the identification of a preferred hybridoma cell clone. The screening proved that the monoclonal IgG class antibody A083, purified by protein A chromatography, binds to ZED1227 when covalently bound to TG2 (Appendix A). As free ZED1227 is expected to be washed away during tissue block preparation and immunofluorescence staining, we expect the signal obtained by A083 to result from the ZED1227-TG2 conjugate. Specificity control experiments are shown in Appendix A.

Anti-TG2. Total transglutaminase 2 was detected with the mouse monoclonal antibody CUB7402 (Thermo Fisher, Waltham, MA, USA), which has been widely used in celiac disease immunofluorescence studies (e.g., [2]).

Anti-gliadin. Detection of gliadin peptide was done with mouse monoclonal antibody A161 (Zedira), which was raised against the synthetic deamidated gliadin-related peptide Lys57-Glu65-[α-gliadin (58-73)] (KLQPFPQPELPYPQPQ). It detects both non-deamidated (native) gliadin and deamidated alpha-gliadin 33 mer, but not gamma-gliadin 26 mer (Zedira).

### 4.3. Single-Color Immunofluorescence Staining

For the single-color immunofluorescence staining, CUB7402, A161, and A083 were used as primary antibodies to detect total TG2, gliadin peptides, and ZED1227-TG2, respectively. Optimal working dilutions were determined for each antibody (1:1000 for CUB7402, 1:25 for A161, and 1:1000 for A083). The immunofluorescence staining protocol included de-paraffinization, high-temperature antigen retrieval (Tris-EDTA buffer, pH 9.0, 121C for 2 min), blocking of endogenous peroxidase activity, incubation with primary antibodies for 1 h, followed by incubation with anti-mouse horseradish peroxidase (HRP) polymer (30 min), and 1:200 diluted CF594-tyramide or CF488-tyramide fluorochromes (Biotium, Fremont, CA, USA). Hematoxylin was used for counterstaining and DPX mounting medium for embedding (Merck Life Science, Darmstadt, Germany). The staining procedure was carried out using an automated staining platform (LabVision Autostainer, ThermoFisher, Waltham, MA, USA).

### 4.4. Dual-Color Immunofluorescence Staining

Dual-color immunofluorescence staining for ZED1227-TG2 and total TG2 was performed with mab A083 and mab CUB7402. Antigen unmasking and blocking of endogenous peroxidase activity were done as described in single-color staining. Primary antibodies and detection reagents were applied to the tissue sections as sequential incubations, employing three washes between the staining rounds. ZED1227-TG2 was detected with red fluorescence (CF594-tyramide) and total TG2 with green fluorescence (CF488-tyramide). Hematoxylin was used to counterstain.

### 4.5. Confocal Microscopy

A Zeiss LSM 780 confocal microscope was used to capture high-resolution Z-stacks of images from the fluorescence-stained slides described above. An oil-immersion 60× Plan-Apo objective was used (numerical aperture 1.4). Spatial calibrations were xy = 67 nm/pixel and z = 200 nm/pixel. Fluorescence emission and excitation filters were matched with the green and red fluorochromes. Stacks of images (2048 × 2048 pixels) were captured and stored as uncompressed .tif image files. A fluorescence microscopy co-localization algorithm was used to highlight the areas (pixels) with statistically significant co-localization (ImageJ/FIJI, [16]).

### 4.6. Digital Image Analysis

Slides were scanned as high-resolution whole-slide images at a resolution of 0.17 µm per pixel (SlideStrider scanner, Jilab Inc., Tampere, Finland) with three alternative illuminations. First, brightfield illumination (for hematoxylin counterstaining) was used to define the tissue boundaries and the focus plane and to provide the histologic context of the biopsy. The second image layer was the scan with green-emitting fluorescence light, and the third was for capturing the deep red-emitting fluorescence. The image layers were saved in the multilayer JPEG2000 image format and viewed with the SlideVantage whole-slide viewing system (Jilab). SlideVantage allows viewing each layer separately or blended to reveal fluorescence co-localization.

For scoring the staining intensity, we developed a quantitative digital image analysis algorithm to measure the mean integrated fluorescence signal intensities of the enterocyte or lamina propria tissue areas defined by the user (Figure 3B,C). Baseline and end-of-study biopsies, placed on the same microscope slide, were measured, and the mean EOS/BL fluorescence intensity ratios were calculated. The fluorescence intensity of the pretreatment biopsy was considered the level of background and was subtracted from the result of the post-treatment biopsy. Pre- and post-treatment biopsies from each patient were placed and stained on the same microscope slide.

### 4.7. Intestinal Organoid Culture

Samples from duodenal biopsies from the Tampere University Hospital (non-celiac patients). Isolation and culture of human epithelial organoids collected from the small intestinal mucosa were carried out as described elsewhere [12]. Briefly, isolated crypts were mixed with Matrigel (BD Biosciences, San Jose, CA, USA) and cultured in WELR500 medium, containing advanced DMEM/F12 medium (Thermo Fisher Scientific, Waltham, MA, USA) with 100 U/mL penicillin and 100 μg/mL streptomycin (Thermo Fisher Scientific), 10 mM HEPES (Thermo Fisher Scientific), 2 mM GlutaMAX (Thermo Fisher Scientific), 1.25 mM N-acetylcysteine (Sigma-Aldrich, Saint Louis, MO, USA), 1 × B27 supplement (Thermo Fisher Scientific), 1 × N2 supplement (Thermo Fisher Scientific), 10 nM gastrin I (Sigma-Aldrich), 10 % WNT3A-conditioned medium, 50 ng/mL EGF (PeproTech, South Korea), 0.2 µM LDN-193189 (Selleckchem, Planegg, Germany), 500 ng/mL R-spondin-1 (PeproTech), 500 nM A83-01 (Tocris, Bristol, UK), 10 µM SB202190 (Sigma Aldrich) and 10 mM Nicotinamide (Sigma-Aldrich).

The organoids were studied as an in vitro model to support our finding that ZED1227 accumulates on the surface of the epithelial cells and that ZED1227 can inhibit TG2 there. They were maintained as 3D Matrigel-embedded cultures and transformed to 2D monolayers for in-situ staining as described [12]. For cell differentiation, WELR500 medium was changed to ELR1 medium (BCM, EGF 50 ng/mL, LDN-193189 0.1 µM ja R-Spondin-1 250 nng/mL) and incubated for 48 h, allowing enteroid cell lineage differentiation (Dotsenko et al., 2023). For intestinal stem cell proliferation, R-spondin-1 concentration was increased to 1000 ng/mL, and media was supplemented with 5 µM CHIR 99021 (Tocris) (WELR1000C medium). RNA was isolated at passage 5 after 3 days in culture, comprising 24 h in WELR500 medium and 48 h in either ELR1 or stem cell proliferation medium. The organoid tissue fragments differentiated on WELR500 medium were picked up from the culture, immersed in Paxgene fixative for 4 h, and processed as paraffin blocks as described for in vivo biopsies.

### 4.8. RNA Isolation and Real-Time Reverse Transcription-Quantitative PCR (RT-qPCR) Analysis

Total RNA was isolated using TRIzol Reagent (Carlsbad, CA, USA), following manufacturer instructions. 500 ng of total RNA was subjected to cDNA synthesis using the iScript cDNA Synthesis Kit (Bio-Rad, Hercules, CA, USA). Real-time PCR reactions were performed with SsoFast EvaGreen Supermix (Bio-Rad, Hercules, CA, USA) and oligos for human TGM2 (forward–5′-TGTGGCACCAAGTACCTGCTCA-3′ and reverse–5′-GCACCTTGATGAGGTTGGACTC-3′), LGR5 (forward–5′-CTGTCGTCTTTTCCTATAACTGGG-3′ and reverse–5′-GAATCTTATAGGCATTCTCACACAC-3′), ALPI (forward–5′-CCAAGTCCTTGGGTCTACCA-3′ and reverse–5′-TCCCCTCAGGTTGTTCTCTG-3′) and GAPDH (forward–5′-GTCTCCTCTGACTTCAACAGCG-3′ and reverse–5′-ACCACCCTGTTGCTGTAGCCAA-3′) in triplicate.

Results presented are calculated as a log2 Fold change compared to the reference sample (WELR500), normalized by housekeeping gene expression (GAPDH) as described [13]. Plot whiskers represent the Standard Error for the mean difference between two independent means.

### 4.9. Transglutaminase In Situ Activity Staining with Biotin-Pentylamine (5-BP) Reagent

The 2D organoids were differentiated in ELR medium for 48 h and subsequently treated either with 0.02 mg/mL, 0.002 mg/mL ZED1227, or DMSO-vehicle diluted in ELR medium for 30 min at 37 °C in a humidified CO_2_-incubator. 5-BP staining was performed as described [12].

### 4.10. Ethics

The biopsies used for the organoids were obtained from the Tampere University Hospital. Written informed consent was given by all participants in compliance with the Declaration of Helsinki. The samples were obtained and processed anonymously. The use of the patient biopsies taken in the CEC-3/CEL study was approved by TUKIJA (dnro 223/06.00.01/2017) and EudraCT (2017-002241-30).

## Figures and Tables

**Figure 1 ijms-24-10815-f001:**
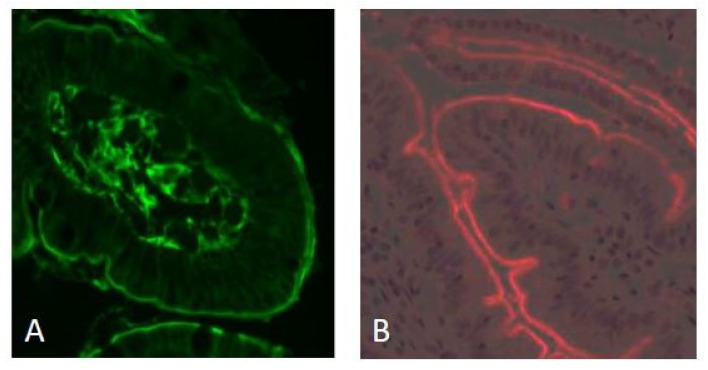
Immunofluorescence staining of transglutaminase 2 (using mab CUB 7402, green fluorescence) and gliadin peptides (mab A161, red fluorescence) in a duodenal biopsy from a CeD patient. The majority of the total TG2 is localized near the basement membrane and in the lamina propria. The luminal surface of the villous epithelium is labeled with equally strong intensity (panel (**A**)). Gliadin peptides (33-mer, non-deamidated, and deamidated) also localize on the luminal surface of the villous epithelium (panel (**B**)). Original magnification: ×200.

**Figure 2 ijms-24-10815-f002:**
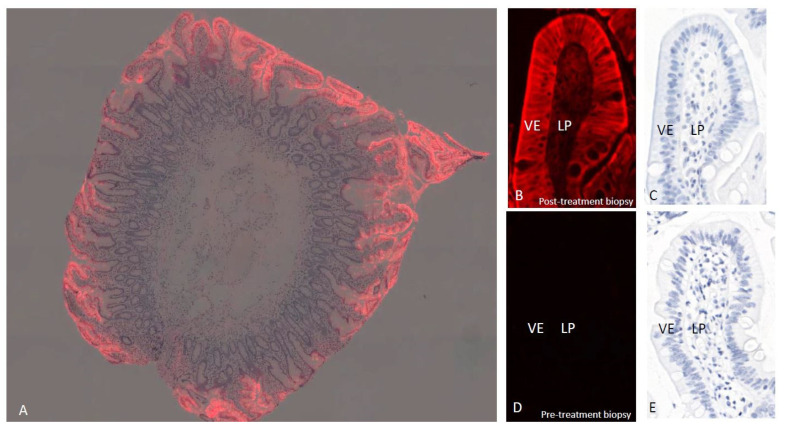
Localization of ZED1227 by mab A083 immunofluorescence (red) in a duodenal biopsy from a ZED1227-treated patient at low magnification (panel (**A**), image blended with hematoxylin counterstain). The signal accumulation in the villous epithelium is evident. At higher magnification, a post-treatment (100 mg/day) biopsy shows the labeling localized mainly in the apical surface of the villous epithelium (VE) (panel (**B**), counterstain in panel (**C**)). Lower-intensity ZED1227-TG2 labeling can be seen in the lamina propria (LP). The fluorescence signal is absent in a pre-treatment biopsy from the same patient (panels (**D**,**E**)).

**Figure 3 ijms-24-10815-f003:**
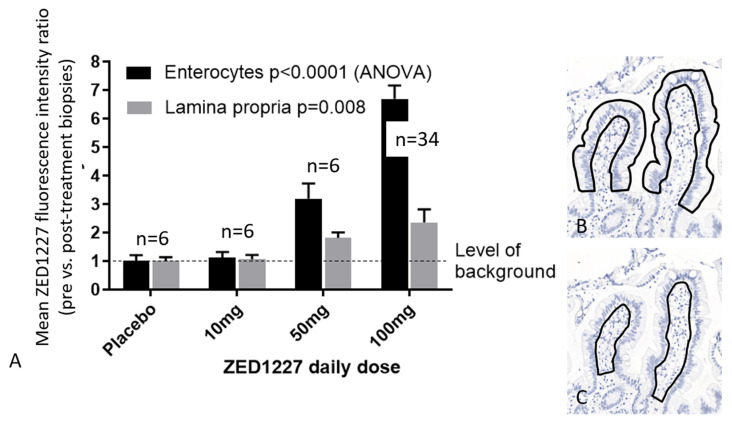
Comparison of quantified ZED1227-TG2 fluorescence intensities in the villous epithelium and lamina propria in the patient groups who received placebo, 10 mg, 50 mg, or 100 mg of ZED1227. The last oral drug dose was given 24 h before endoscopy and biopsy sampling (panel (**A**)). Panels (**B**,**C**) illustrate the regions drawn to define the epithelial and lamina propria measurement areas. Shown are the average fluorescence intensity ratios (pre- vs. post-treatment biopsies of each patient stained on the same slide). Original magnification in panels B and C, ×100.

**Figure 4 ijms-24-10815-f004:**
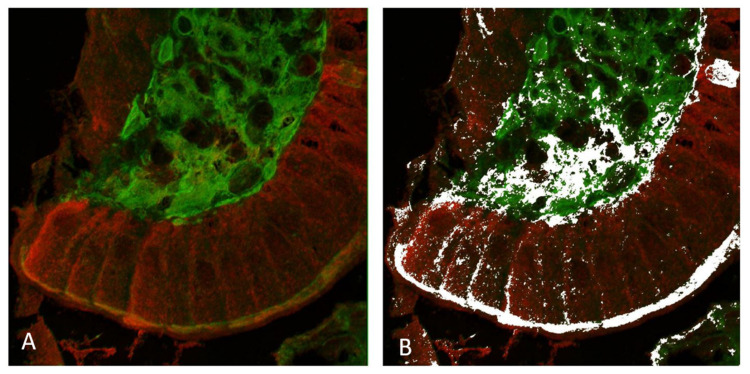
Confocal microscopy imaging of total TG2 (mab CUB 7402, in green) and ZED1227 (mab A083, in red) by double immunofluorescence staining (panel (**A**)). White pixels in panel (**B**) show the areas with statistically significant co-localization defined by the ImageJ/FIJI image analysis co-localization algorithm. Original magnification: ×600.

**Figure 5 ijms-24-10815-f005:**
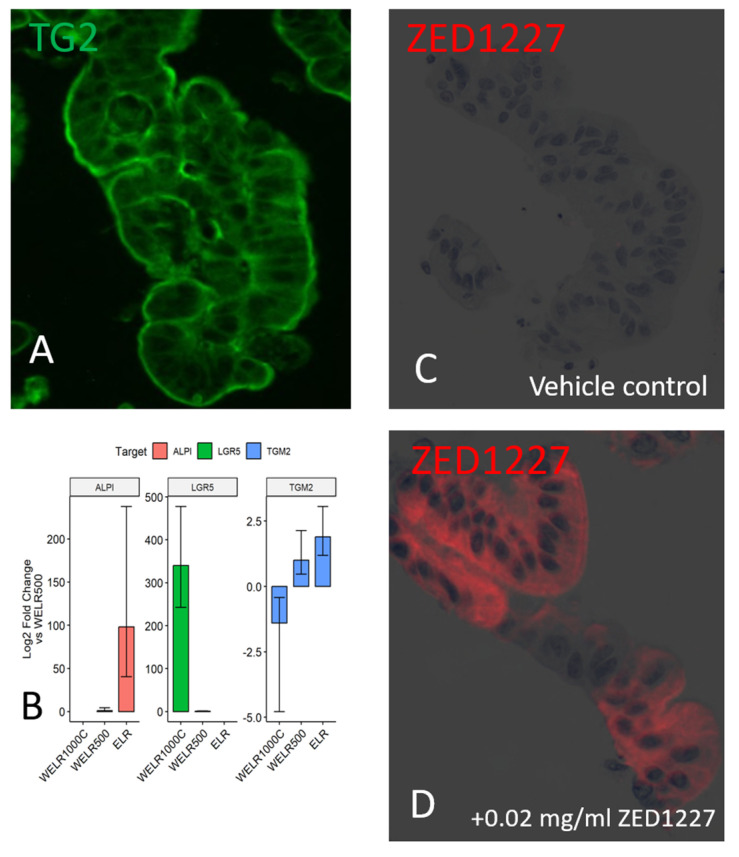
Panel (**A**) shows the immunofluorescence localization of total TG2 in a lamina-propria- free intestinal human epithelial organoid. Panel (**B**) shows the expression level of TG2 mRNA, as determined by RT–qPCR. The mRNA level is increased along with villous differentiation (using the ELR medium). The increase parallels with that of the mature enterocyte marker (ALPI), while the mRNA level of the stem cell marker LGR5 is decreased. Plots with whiskers represent the mean and standard error (*n* = 2). Panels (**C**,**D**) demonstrate the localization of ZED1227 in the epithelial organoid cells before (panel (**C**)) and after adding 0.02 mg/mL of ZED1227 in the culture medium (red fluorescence).

**Figure 6 ijms-24-10815-f006:**
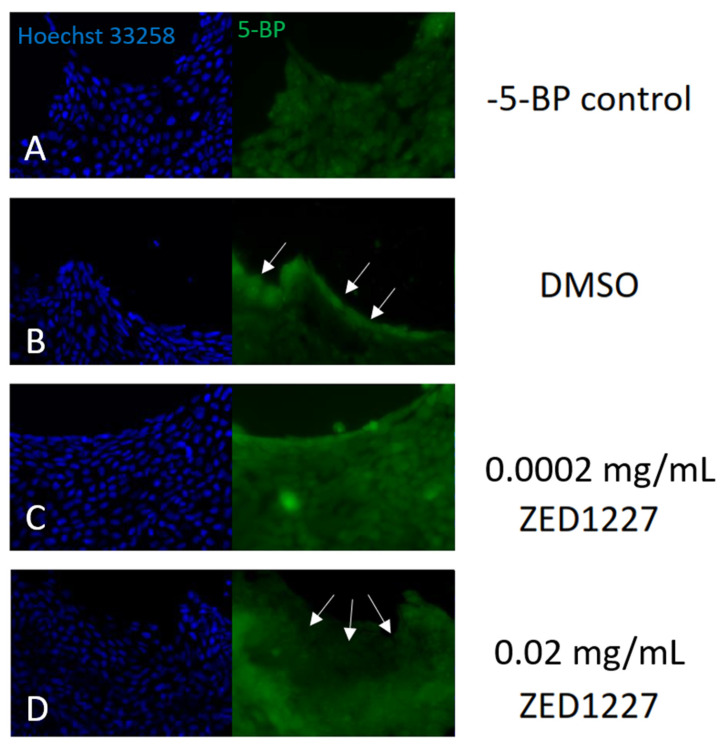
TG2 activity is demonstrated with 5-biotin pentylamine labeling in a cultured human epithelial organoid. The TG2 inhibitor ZED1227 (0.02 mg/mL) caused a reduction of 5-BP staining intensity in the apical epithelial surface compared to the DMSO control (arrows). Panel (**A**) (negative control); panel (**B**) (DMSO control; no TG2 inhibitor added); panels (**C**,**D**) (with ZED1227 0.002 mg/mL and 0.02 mg/mL added, respectively).

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
