# Peer review of "The Oral Transglutaminase 2 Inhibitor ZED1227 Accumulates in the Villous Enterocytes in Celiac Disease Patients during Gluten Challenge and Drug Treatment"

_ijms, 2023, doi:10.3390/ijms241310815_

Round 1

Reviewer 1 Report

This is an interesting article highlighting new insights into the localization of transglutaminase 2 activity in celiac disease (CD) in conjunction with the inhibitory activity of a new drug candidate for CD. The article is well written and clear, only some points should be improved.

In Materials and methods, Patients and biopsies section, a description of how many patients/samples were tested for each analysis should be added.

In the Intestinal organoid culture section, a description of the various stages of organoid testing for the various analyses should be added. The description of RT-qPCR should be improved with an indication of when the RNA was analyzed, in how many samples and why an analysis of treated and not treated (with ZED1227) organoids was not performed.

It seems that organoids were estabilished only from “non-celiac patients” (which patients? healthy donors?) and not from celiac patients, even though the manuscript deals with biopsies from patients with CD; if so, it should be important for the authors to explain why, because a comparison between healthy controls and CD patients would also be helpful.

The text lacks the reference to figure 6: it should be inserted in Results, organoid model.

The number of patients is also missing in figure 3A.

Author Response

This is an interesting article highlighting new insights into the localization of transglutaminase 2 activity in celiac disease (CD) in conjunction with the inhibitory activity of a new drug candidate for CD. The article is well written and clear, only some points should be improved.

In Materials and methods, Patients and biopsies section, a description of how many patients/samples were tested for each analysis should be added.

REPLY: The numbers of patient biopsies studied for ZED1227 immunofluorescence staining intensity analysis (Fig 3.) were:

Placebo group: n=6

10 mg and 50 mg groups both  n=6

100 mg group n=34 (all biopsies available from this group).

--- The numbers are now added above the bars in Fig. 3A.

In the Intestinal organoid culture section, a description of the various stages of organoid testing for the various analyses should be added.

It seems that organoids were estabilished only from “non-celiac patients” (which patients? healthy donors?)  and not from celiac patients, even though the manuscript deals with biopsies from patients with CD; if so, it should be important for the authors to explain why, because a comparison between healthy controls and CD patients would also be helpful.

REPLY: The text below is now added to the Materials and Methods

“Organoids were utilized as an in vitro epithelial cell model to support our finding that ZED1227 accumulates on the surface of epithelial cells and that ZED1227 can inhibit TG2 there. Organoids were established from 2 unrelated healthy donors. RNA was isolated at passage 5 after 3 days in culture, comprising 24 hours in WELR500 medium and 48 hours in either ELR1 medium or stem cell proliferation medium.  The organoids differentiated on WELR500 were used TG2 and ZED1227 immunofluorescence experiments.“

---

The description of RT-qPCR should be improved with an indication of when the RNA was analyzed, in how many samples and why an analysis of treated and not treated (with ZED1227) organoids was not performed.

REPLY: The RT-qPCR experiment showed that TG2 mRNA and protein on the enterocyte surface are synthesized by the enterocytes, because the organoids have no lamina propria. Administration of ZED1227 did not alter the mRNA expression of TG2 (data not shown).

(now added in the text).

We tried to keep the method description of RT-qPCR short, because the method has been described in detail by us in the reference #12.

Reviewer 2 Report

Isola et al. demonstrate the target engagement of the transglutaminase 2 irreversible inhibitor ZED1227 by performing immunohistochemical analysis of biopsies obtained from patients undergoing therapeutic treatment of celiac disease with the inhibitor.  an antibody recognizing the covalent enzyme-inhibitor complex was employed, which was raised by immunization with the inhibitor-thiol conjugate as developed within this study. Besides obtaining proof of dose-dependent targeting of transglutaminase 2 in the diseased intestine, the results of this study have revealed that the enzyme is mainly active in the villous epithelial cells and to a minor extent in the lamina propria, in contrast to previous assumptions. The results were corroborated in human epithelial organoid. Therefore, this work is highly recommended for publication. The following points should be considered during revision:

The short first paragraph of the introduction should be extended for a somewhat more detailed concise outline of the pathogenesis of celiac disease, which will be for the less initiated reader. Furthermore, a Figure showing the layers of the intestinal mucosa would be helpful for those readers who less familiar with the microscopic anatomy of the intestine.

I would like to suggest including a figure showing the chemical structure and mechanism of inhibition (i.e. the covalent adduct with the active-site cysteine residue) of ZED1227.

Page 2: The wording “the Michael acceptor warhead was replaced” is slightly misleading. Rather it should be stated that ZED1227 was reacted with homocysteine (Why was used this thiol?) to form the thia-Michael adduct resembling the covalent enzyme-inhibitor complex. How was the adduct conjugated to KLH?

Page 3: What is DPX?

Page 4: First sentence reads awkward and needs revision

The authors state that the ZED1227-transglutaminase 2 complex is located in the cytoplasm, which I think is a very exciting finding. Potential mechanisms that lead to intracellular activation of transglutaminase 2 in the context of celiac disease should be discussed.

Figure 1: The tissue material of the section should be specified. Explicit caption for panel B is missing.

Figure 2_ Reference to panel C is missing.

Figure 3: “quantitate” should be replaced by “quantified”

Figure 5: Axis titles and legend in subfigure B are difficult to read. Caption text: “0.02 mg/ml TG2” should probably read “0.02 mg/ml ZED1227”

Reference 6: Initial of last author seems to be missing

Reference 10: Author names are missing

Title page: Order of first and surnames should be changed. Corresponding author is not assigned.

Quality of language seems to be sufficient.

Author Response

This study appears to be well designed and rigorously conducted. Its results are original and clearly presented; the discussion, while concise, is fair and thorough. This reviewer has only a few minor remarks that should be addressed.

  1. Abstract: The expression "villous enterocytes of the small bowel mucosa" is probably redundant; in fact, the enterocytes are by definition "of the small bowel mucosa".

REPLY: Corrected as suggested.

Also: It would be useful to the reader to add, between Background and the Materials and Methods, a brief paragraph on "Aims". 

REPLY: The abstract had a 250 word limit. Despite the limit, we modified the text as follows:

“Our aim was to study the accumulation of ZED1227 in vivo after oral administration of the drug.”

  1. In the section "Inhibition of TG2 by ZED1227 in a lamina propria-free intestinal organoid model" on the 9th row, it is quoted for the second time "Fig. 5A" while the results presented are actually those found in Fig. 5 D. Please correct.

REPLY: Corrected.

III. In Fig. 3, what is meant by "EOS" in the Y label of the graph?

+

REPLY: We apologize for not explaining the annotation: “EOS” means “end-of-study” , i.e. the biopsies taken after gluten challenge. Now corrected in the revised Figure 3.

  1. In Fig. 5, "Panels C and D demonstrate the localization of ZED1227 in the epithelial organoid cells after adding 0.02 mg/ml of TG2 in the culture (red fluorescence)." it would be better phrased as "...organoid cells before (C) and after (D) adding 0.02....

REPLY: Corrected as suggested.

  1. Figure 6 is not at all commented or even referenced in the text. Either delete it or insert proper call-out in the text. The text lacks the reference to figure 6: it should be inserted in Results, organoid model.

REPLY: reference to Fig. 6 is now added in the Results, here:

“The effect of ZED1227 on the TG2 activity was studied after adding the substrate 5-biotin pentylamine (5-BP). Staining intensity on the epithelial cell surface, reflecting TG2 activity, was decreased dose-dependently compared to buffer control after treatment with 0.002 or 0.02 mg/ml of ZED1227 (Fig. 6.).

The number of patients is also missing in figure 3A.

REPLY: Now added in Fig 3A

Reviewer 3 Report

 This study appears to be well designed and rigorously conducted. Its results are original and clearly presented; the discussion, while concise, is fair and thorough. This reviewer has only a few minor remarks that should be addressed.

I. Abstract: The expression "villous enterocytes of the small bowel mucosa" is probably redundant; in fact, the enterocytes are by definition "of the small bowel mucosa".

Also: It would be useful to the reader to add, between Background and the Materials and Methods, a brief paragraph on "Aims". 

II. In the section "Inhibition of TG2 by ZED1227 in a lamina propria-free intestinal organoid model" on the 9th row, it is quoted for the second time "Fig. 5A" while the results presented are actually those found in Fig. 5 D. Please correct.

III. In Fig. 3, what is meant by "EOS" in the Y label of the graph?

IV. In Fig. 5, "Panels C and D demonstrate the localization of ZED1227 in the epithelial organoid cells after adding 0.02 mg/ml of TG2 in the culture (red fluorescence)." it would be better phrased as "...organoid cells before (C) and after (D) adding 0.02....

V. Figure 6 is not at all commented or even referenced in the text. Either delete it or insert proper call-out in the text.

Author Response

Isola et al. demonstrate the target engagement of the transglutaminase 2 irreversible inhibitor

ZED1227 by performing immunohistochemical analysis of biopsies obtained from patients

undergoing therapeutic treatment of celiac disease with the inhibitor.  an antibody recognizing the

covalent enzyme-inhibitor complex was employed, which was raised by immunization with the

inhibitor-thiol conjugate as developed within this study. Besides obtaining proof of dose-dependent

targeting of transglutaminase 2 in the diseased intestine, the results of this study have revealed that

the enzyme is mainly active in the villous epithelial cells and to a minor extent in the lamina propria,

in contrast to previous assumptions. The results were corroborated in human epithelial organoid.

Therefore, this work is highly recommended for publication. The following points should be

considered during revision:

The short first paragraph of the introduction should be extended for a somewhat more detailed

concise outline of the pathogenesis of celiac disease, which will be for the less initiated reader.

REPLY: Done as suggested. See the 1st paragraph of the Introduction of the revised manuscript.

Furthermore, a Figure showing the layers of the intestinal mucosa would be helpful for those readers

who less familiar with the microscopic anatomy of the intestine.

REPLY: We prepared a new figure (Supplement Fig 3.) of the duodenal mucosa and added its reference on page 7/13. 

I would like to suggest including a figure showing the chemical structure and mechanism of

inhibition (i.e. the covalent adduct with the active-site cysteine residue) of ZED1227.

REPLY: In order to keep the article short and coherent, we refer to Buchold C et al. (ref #10), which shows the chemical structure of ZED1227.

Page 2: The wording “the Michael acceptor warhead was replaced” is slightly misleading. Rather it

should be stated that ZED1227 was reacted with homocysteine (Why was used this thiol?) to form

the thia-Michael adduct resembling the covalent enzyme-inhibitor complex. How was the adduct

conjugated to KLH?

REPLY: The use of maleimide activated KLH is now described on page 3/13.

Page 3: What is DPX?

REPLY: DPX is a xylene-based mounting medium for histology (from Merck) we used in this study. Now explained in M&M.

Page 4: First sentence reads awkward and needs revision

REPLY: Revised as:  “Samples from duodenal biopsies were derived from from the Tampere University Hospital (non-celiac patients).   

----------------------

The authors state that the ZED1227-transglutaminase 2 complex is located in the cytoplasm, which I

think is a very exciting finding. Potential mechanisms that lead to intracellular activation of

transglutaminase 2 in the context of celiac disease should be discussed.

REPLY: Our interpretation from confocal microscopy images was that TG2 and ZED1227 were cytoplasmic. It is possible that the strongest signal of ZED1227 and TG2 are actually in the brush border, namely on the villus epithelium.  Now explained on page 7/13.

Figure 1: The tissue material of the section should be specified. Explicit caption for panel B is missing.

REPLY:  Now explained in the Figure legend. Caption to panel B added.

Figure 2_ Reference to panel C is missing.

REPLY: Now added in the Figure Legend.

Figure 3: “quantitate” should be replaced by “quantified”

REPLY: Corrected.

Figure 5: Axis titles and legend in subfigure B are difficult to read. Caption text: “0.02 mg/ml TG2”

should probably read “0.02 mg/ml ZED1227”

REPLY: Corrected.

Reference 6: Initial of last author seems to be missing

REPLY: Corrected.

Reference 10: Author names are missing

REPLY: Corrected.

Title page: Order of first and surnames should be changed. Corresponding author is not assigned.

REPLY: Corrected. The corresponding author is prof. Jorma Isola, now shown with an asterisk.